# The Role of Odor-Evoked Memory in Psychological and Physiological Health

**DOI:** 10.3390/brainsci6030022

**Published:** 2016-07-19

**Authors:** Rachel S. Herz

**Affiliations:** Department of Psychiatry and Human Behavior, Brown University Medical School, 190 Thayer St., Providence, RI 02912, USA; Rachel_herz@brown.edu; Tel.: +1-401-863-9576

**Keywords:** odor, emotion, psychology, autobiographical-memory, immune-response, stress physiology, aromatherapy, gender, personality

## Abstract

This article discusses the special features of odor-evoked memory and the current state-of-the-art in odor-evoked memory research to show how these unique experiences may be able to influence and benefit psychological and physiological health. A review of the literature leads to the conclusion that odors that evoke positive autobiographical memories have the potential to increase positive emotions, decrease negative mood states, disrupt cravings, and reduce physiological indices of stress, including systemic markers of inflammation. Olfactory perception factors and individual difference characteristics that would need to be considered in therapeutic applications of odor-evoked-memory are also discussed. This article illustrates how through the experimentally validated mechanisms of odor-associative learning and the privileged neuroanatomical relationship that exists between olfaction and the neural substrates of emotion, odors can be harnessed to induce emotional and physiological responses that can improve human health and wellbeing.

## 1. Introduction

Odor memory is a central feature of olfactory cognition, and can be divided into two distinct cognitive-perceptual processes [1,2,3]. One is the ability to recognize and remember whether one has smelled an odor before. This form of odor memory is similar to recognizing other sensory semantic cues, such as knowing that a particular sound signifies your dog’s bark. Odor recognition and identification, and in particular its failure, is a critical factor in human health. Data from a representative sample of over 3000 US community dwelling adults 57 years and older, recently revealed that those who were dysfunctional at odor identification were four times more likely to die within a five year period than their same age peers with normal olfactory abilities [4]. Impairment of odor identification is also a hallmark early symptom and predictor of several neurological disorders, most notably Alzheimer’s disease and Parkinson’s disease. For recent reviews of these topics see Doty [5] and Velayudhan [6]. The second type of odor memory is odor-evoked memory—autobiographical memories and associations that are triggered by odors. This type of memory has not been directly examined for the ways in which it may be involved in human health, and an effort to do so is the subject of the present article.

Odor-evoked memory or the “Proust phenomenon” [7] from the eponymous literary anecdote where Marcel Proust took a bite of madeleine biscuit that had been dipped in Linden tea and was suddenly transported to a long forgotten moment in his childhood [8], occurs when an odor triggers the recollection of a meaningful past personal episode. Odor-evoked memories possess several characteristics that distinguish them from memories evoked by stimuli perceived through other sensory modalities (for reviews see [2,9]. Notably, odor-evoked memories have been shown to be more rare, less frequently thought about [10,11,12], and from an earlier time in life, specifically clustered in the first decade [10,12,13] compared with memories evoked by verbal or visual stimuli. 

The most distinctive characteristics of odor-evoked memories, however, and why they are important to human health and wellbeing is that they evoke more emotional and evocative recollections than memories triggered by any other cue. Numerous studies have now shown that autobiographical memories triggered by odors feel much more emotional, activate the neurolobiological substrates of emotional processing, and that people are more brought to the original time and place of their memories compared to when the same events are recalled through other modalities [12,14,15,16,17,18,19,20]. 

Odor-evoked memories are exceptionally viscerally involving because the neuroanatomy of olfaction has a privileged and unique connection to the neural substrates of emotion and associative learning. The primary olfactory cortex includes the amygdala, which processes emotional experience and emotional memory, as well as the hippocampus, which is involved in associative learning [21,22,23,24]. Thus, the mere act of smelling activates the amydala-hippocampal complex. Moreover, during the process of recollecting an odor-evoked autobiographical memory, the amygdala is more activated than when similar odors that do not evoke a memory are smelled [18]. Additionally, the secondary olfactory cortex (the orbitofrontal cortex) is the cortical area responsible for assigning affective value to stimuli and for determining the reinforcement value of stimuli in general [25,26]. Finally, unlike other sensory systems where neural processing is directly integrated in the thalamus prior to processing in the cortex [27], olfactory information is indirectly processed through the medio-dorsal nucleus of the thalamus which appears to play a role in modulating attention to odors [27,28,29], and the primary processing of odors occurs instead in the amygdala-hippocampal complex of the limbic system. None of our other senses have this level of targeted connection with the areas of the brain that process emotion, associative learning, and memory. 

The present review will explain how through their uniquely emotional and evocative properties odor-evoked memories alter mood and induce the physical correlates of various emotional states, and thus that they are important to, and can be used for, ameliorating psychological and physical health. The following two sections discuss recent research where odor-evoked memories have been found to affect emotional and physiological responses and the implications these findings have for human health. Olfactory factors and individual difference characteristics that would need to be considered in any therapeutic application of odor-evoked are then examined. The paper concludes with a discussion of the mechanisms through which odor-evoked memories exert their positive effects and offers some directions for future research that are now needed in order to mine the therapeutic potential of odor-evoked memory. Table 1 provides a synopsis of the methods and results of the relevant odor-evoked memory studies reviewed here.

## 2. Odor-Evoked Memory and Psychological Responses

Positive moods and emotions are known to be beneficial for psychological health [30] and odor-evoked memories have been shown to be more positive than memories elicited by other cues. Willander and Larsson [20] compared memories evoked by odors, verbal labels, and odors + labels in healthy older adults and found that memories evoked by odors were significantly more pleasant than memories evoked by labels. Additionally, Arshamian and colleagues [14] found that among male and female adults (age range 20–28 years) autobiographical memories elicited by odors were more emotionally intense and positive than memories cued by verbal labels for the same odors. Arshamian et al. [14] also showed that odor-evoked memories were associated with more activity in the temporal gyrus and temporal pole than verbally evoked memories. The temporal poles have been linked to the processing of pleasant memories [31], therefore Arshamian et al. [14] speculated that increased temporal pole activity may underlie the superior positively of odor-evoked-memories. 

Pleasant autobiographical memories in general can induce positive mood, and invoking such memories has been used as a therapeutic technique to repair emotional distress in various clinical conditions [39,40]. Since odors elicit more emotional memories than other types of stimuli, and because odor-evoked memories tend to be positive, odors may be especially helpful for enhancing mood states. Indeed, Matsunaga et al. [36] found that when men and women aged 21–38 were presented with the scent of a perfume that was personally reminiscent for them, more positive emotion, increased moods of comfort and happiness, and a decrease in anxiety were experienced compared to when a pleasant fragrance that did not elicit personal memories was presented. 

Beyond the specificity of memory, nostalgia—reflecting upon one’s personal past—has been shown to have many beneficial psychological consequences. Engaging in nostalgic reminiscence increases positive affect, bolsters self-esteem, strengthens the connection between one’s past and present, produces feelings of social connectedness, elevates optimism, and infuses life with meaning (see [32] for a review). Recent research suggests that odor-evoked memories may be especially nostalgic triggers.

Matsunaga et al. [37] found that among young adults, odors that evoked personal memories elicited approximately 6.5 times more feelings of nostalgia than odors that did not evoke a specific past event. Moreover, a comparison of music-evoked nostalgia with odor-evoked nostalgia showed that odors elicited more than twice as many nostalgic reveries as musical excepts did [32,41]. This is an important finding since music is renowned as an emotionally evocative stimulus and it is often reported that music evokes poignant memories [42]. Further support for the nostalgic power of odors was shown by Miles and Bernsten [13] who presented 12 familiar stimuli (e.g., mustard, coffee) in verbal, visual and olfactory formats to a large sample of female college students and compared various qualitative dimensions of memory. They found that memories evoked by odors were judged as more relevant to one’s life story than memories elicited by verbal or visual cues. 

Reid et al. [32] recently performed an in-depth analysis of odor-evoked nostalgia with college students using 12 common scents and demonstrated that, consistent with the beneficial effects of nostalgia in general, high levels of odor-evoked nostalgia were associated with high levels of positive affect, self-esteem, self and social connectedness, optimism, and life meaning. Reid et al. [32] also found that nostalgic scents elicited three times more positive emotions than negative emotions, and that twice as many pleasant emotions were experienced during odor-evoked nostalgia than what had been previously found in a study examining music-evoked nostalgia [41]. Thus, as with the marked positivity of odor-evoked memories, feelings of nostalgia that are elicited by odors are more pleasant than nostalgic reminiscences elicited by other sensory stimuli. 

It should be noted that odor-evoked memories can also elicit unpleasant emotions. Indeed they can be exceptionally potent triggers in post-traumatic stress disorder (PTSD). Case studies of PTSD have documented how odors that are associated with a traumatic event for a given individual (e.g., diesel, after-shave) evoke intense PTSD flashbacks which persist for decades, and do not extinguish with time [34]. Moreover, in a laboratory investigation of aversive memories it was found that among healthy college students who were asked to recall scenes from a traumatic documentary, which one week earlier had been paired with either an olfactory, visual or auditory cue, the olfactory cue led to more arousing, detailed and unpleasant memories of the documentary than the auditory cue did [33]. These findings caution that depending upon an individual’s past experience with an odor, the emotional states and responses that it elicits can be very negative. 

## 3. Odor-Evoked Memory and Physiological Responses

Positive emotional states have generally been shown to improve physical health and longevity [43]. Therefore eliciting positive emotions through odor-evoked memory should have favorable health outcomes. Considerable empirical research has demonstrated that through odor-associative learning mechanisms—when via evaluative conditioning an odor takes on the meaning of an event to which it has been paired—odor perception and behavior in the presence of these odors can be reliably altered (see [44,45] for reviews). Consistent with the principles of odor-associative learning [46], when an odor has become a proxy for the emotional properties of an event to which it is connected it can consequently produce downstream effects on physiology [47]. For example, an odor that evokes an exhilarating past personal episode can make one’s heart race and trigger a jolt of adrenalin—the physiological sequelae of the exhilarating emotional memory. Additionally, the direct connection between olfactory processing and the amygdala-hippocampal complex can make the emotions elicited by odor-evoked memories arise immediately upon perception. Indeed, emotional primacy—when affect is elicited before a cognitive understanding of why the emotions produced has occurred—is another unique characteristic of odor-evoked memory [48].

Following the tenants of odor-associative learning, it has been experimentally demonstrated that odors that evoke positive emotional memories can have beneficial effects on physiological parameters of stress. In the study by Matsunaga et al. [36], pleasant odor-evoked autobiographical memories were also accompanied by a decrease in HR and an increase in SC, and these changes were correlated with self-reports of a happy mood and lower stress levels. Correspondingly, the authors surmised that the positive emotions elicited by the odor-evoked memories were responsible for these positive physiological responses. Masaoka, Sugiyama, Katayama, Kashiwagi and Homma [35] further showed that when men and women ranging in age from 29 to 50 years were presented with a self-selected odor that evoked an autobiographical memory odor and two control fragrances, a pleasant odor (rose) and a neutral odor (chamomile) that did not elicit personally meaningful recollections, the autobiographical memory odor promoted deeper, slower and more relaxed breathing compared to odors that did not evoke a memory. 

Slow, deep breathing is associated with relaxation and may stimulate whole-brain synchronization as it is seen in slow-wave sleep and meditation [49], both of which have neurological benefits. Masaoka et al. [35] therefore speculated that the respiratory changes that accompany the experience of pleasant odor-evoked memories may help lower stress and increase comfort—as feeling comfort was also reported by participants in this study. Importantly, the evocativeness of the autobiographical memories triggered by odors and depth of breath were found to be greatest among individuals who scored high in trait anxiety, implying that the stress reducing benefits of odor-evoked memories may be strongest for those with the most need for it. 

Chronic stress is correlated with a host of negative psychological and physiological consequences ranging from depression and memory-loss to stroke and cancer (for a few reviews see [50,51,52]). A positive and calm mood is an antidote to stress, though when one is in a state of stress, pleasant moods are very hard to experience and maintain. Nevertheless, many people intuitively know that odors have the potential to elicit comfort. The act of “comfort smelling” where a person sniffs a garment worn by a loved one they are separated from has been demonstrated to conjure the feelings of love, support, and comfort representative of the absent person [53,54].

Following from the intuitive behavior of comfort smelling, the deliberate activation of comforting odor-evoked-memories by smelling an article of clothing, a fragrance, or any specific odor that for a given individual evokes soothing feelings of relaxation, may reduce stress. Extrapolating from this I have further suggested that, when not in a state of physical hunger, merely smelling the aromas of “comfort foods”—foods which are especially sought after during times of difficulty as a function of their connection to memories of home, loved ones, and security—may be able to produce emotional succor and reduce stress without the negative consequences of over-eating [55]. 

Using odor-evoked memory to reduce stress can minimize the negative consequences that often accompany self-medicating methods such as excessive drug, alcohol and food intake, and may even be helpful in reducing cravings for various substances of abuse. Sayette and Parrot [38] found that smelling odors decreased the urge for cigarettes among chronic smokers, and odors that elicited autobiographical memories trended towards being most effective. Several studies have now also reported that, at least in laboratory settings, odors can disrupt cravings for highly desirable calorie dense foods [56,57]. These studies on food craving did not assess whether the odors involved evoked personal memories. Nevertheless, it is likely that the mechanisms through which odors reduced food cravings were memory based. For example, aromas connected to rich and filling foods could have elicited feelings of satiation, other odors may have triggered reminders of dieting intentions, and memories triggered by odors may have distracted individuals from their immediate cravings and thus disrupted capitulating to these urges [55]. These possibilities now need to be addressed in further research.

Neuroimaging studies investigating brain-immune interactions have revealed that prefrontal regions, such as the ventromedial prefrontal cortex (vmPFC) and the orbitofrontal cortex (OFC), regulate peripheral immune activities such as the proportion of natural killer cells among peripheral circulating lymphocytes [58,59]. Since the OFC is the secondary olfactory cortex it follows that odors may be able to exert a direct influence on immune responses. 

Recent research that has examined physiological parameters associated with odor-evoked memory has found that odor-evoked memories may decrease systemic inflammation. Matsunaga et al. [36] demonstrated that pleasant odor-evoked memories were accompanied by a decrease in plasma levels of the inflammatory cytokine interleukin 2 (IL-2). IL-2 is normally produced by T-lymphocytes during an immune response. A decrease in peripheral IL-2 therefore implies an inhibition of inflammation. As such, Matsunaga et al. [36] suggested that the positive feelings accompanying odor-evoked memories were able to decrease levels of peripherally circulating pro-inflammatory cytokines. 

To further investigate the relationship between odor-evoked memory and brain-immune interactions, Matsunaga and colleagues [37] conducted an experiment where healthy men and women underwent positron emission tomography (PET) while they were exposed to a perfume that they had selected as eliciting an autobiographical memory and a pleasant unfamiliar perfume that did not have any personal significance for them (control fragrance) and plasma concentration of several inflammatory cytokines were measured. Results showed that the self-selected odors elicited more positive emotions and autobiographical memories than the control odor, and importantly that levels of the peripheral proinflammatory cytokines, tumor necrosis factor-α (TNF-α) and interferon-γ (IFN-γ), were significantly reduced after experiencing an odor that evoked an autobiographical memory. IL-2 was measured in this study but was not found to vary as a function of odor exposure condition, unlike Matsunaga group’s previous study [36]. The PET data further revealed that the medial orbitofrontal cortex (mOFC) and precuneus/posterior cingulate cortex (PCC) were significantly activated during experiences of odor-evoked autobiographic memory, and correlational analyses indicated that activation of the mOFC and precuneus/PCC were negatively associated with IFN-γ concentration. 

Increased levels of proinflammatory cytokines, such as TNF-α and IFN-γ, can induce depressive symptoms [60]. Moreover, recent studies in psychoneuroimmunology have shown that emotional experiences can modulate the secretion of proinflammatory cytokines from immune cells, and that psychological stressors, such as anxiety, can promote secretion of proinflammatory cytokines [61]. It has also been demonstrated that happy feelings can suppress the secretion of proinflammatory cytokines and that they have favorable effects on health and wellbeing [62]. The results from Matsunaga’s group [37], therefore, suggest that a brain-immune interaction occurs during the experience of emotionally positive odor-evoked autobiographical memories such that the precuneus/PCC and mOFC, which regulate the secretion of peripheral proinflammatory cytokines, produces a decrease in peripheral pro-inflammatory responses. 

Excessive inflammation has been implicated as the precursor to nearly every deleterious physical condition [63]. If odor-evoked-memories have the ability to reduce inflammation, even if the effects are minimal, it would be beneficial. A caveat to over-reaching on this conclusion, however, is that blood tests in the Matsunaga experiments [36,37] were taken within two minutes of odor exposure so it is not known how long the positive effects from odor-evoked memory on immune function would last. Cautiously, one might posit that as long as a positive emotional state were maintained from an odor-evoked memory a reduction in inflammatory cytokines may persist. Conversely, based on the extant research on stress and inflammation and studies demonstrating the link between olfaction and brain-immune interactions, odor-evoked memories that elicit substantial distress, such as which occur in PTSD, may have the potential to increase inflammation, though this has not yet been empirically assessed. 

## 4. Issues for Consideration

### 4.1. Odor Perception

Olfactory adaptation and habituation are perceptual factors that are important to consider when odors are frequently experienced. When an odor is continuously or repeatedly smelled it loses its ability to be perceived. After constant exposure to an odor for more than 20 min sensitivity is greatly reduced and when one is exposed to the same odor on a daily basis olfactory sensitivity to that odor dramatically diminishes [64,65]. Therefore, limited exposure to odors that elicit autobiographical memories and use of multiple odors with this capacity is advised when using odors that evoke memories for therapeutic purposes.

There is also some evidence that odors that stimulate the trigeminal nerve (e.g., mint) may evoke less pleasant memories than odors which do not (e.g., vanilla), and that memories evoked by trigeminally activing odors are clearer than memories evoked by nontrigeminal odors [66]. Therefore, specific odorant qualities may also be important to consider in treatment applications.

Another important issue is that olfactory cognition is unlike cognition mediated through our other senses in that it is extremely resistant to retroactive interference [67,68]. That is, the first association formed to an odor typically remains tied to that odor despite multiple future experiences of the same odor in different contexts. This is an advantage in that the same odor can be used to repeatedly recapture a specific past event—so long as issues of odor habituation have been mitigated—but also means that negative odor-evoked memories, such as in PTSD, are extremely hard to extinguish. 

In order for an odor to be therapeutically effective it must be both associated to, and perceived as, possessing specific positive emotional qualities. That is, the scent of lavender may induce positive feelings of relaxation [69,70], but only if lavender scent is known to be and associated to pleasant calming emotions for a given individual. If a certain scent is not known to be “lavender”, if it is disliked, or if it believed to be stimulating it may instead increase physiological indices of arousal. For example, Campenni, Crawley and Meier [71] demonstrated that when lavender odor was presented un-named and with the description that it was “stimulating” it increased HR, and when the same odor (un-named) was presented with the description that it was “relaxing” it decreased HR. Moreover, regardless of one’s past experience, odor perception is variable. We have recently shown that olfactory sensitivity fluctuates with circadian phase and is most acute in the late afternoon and worst in the early morning [72]. This suggests that the time of day when odors are presented in therapeutic applications would also be important to consider. 

### 4.2. Experience and Genetics

The occurrence of personally meaningful odor-evoked memories is relatively rare. Willander and Larsson [20] suggested that 16% of autobiographical memories are elicited by odors, and by their nature odor-evoked memories involve idiosyncratic experiences. Therefore, even if a scent is generally denoted as positive, the odor-evoked memory that a given individual experiences to it may not be. For instance, the scent of a popular cologne though evocative of romance for many, can trigger PTSD flashbacks for a rape victim [34]. Culture also plays an essential role in past experience with an odor as it provides the framework upon which odor learning takes place, and it has been shown that there are widely differing hedonic responses to “everyday” odors across cultures [73]. Thus, in addition to idiosyncratic experience, the predictive emotional effects of a certain odor evoking a memory are limited to the culture(s) in which the acquired associations fit the expected responses [47]. 

It is also now known that each of us, unless we are an identical twin, possess a unique complement of functional olfactory receptors [74,75,76]. This means that there are subtle differences in how we all perceive odors. The olfactory receptors that we possess reflect which genes are expressed as functional receptors and how many copies of a specific receptor a person may have. For example, people who despise the aroma of cilantro are missing the gene for detecting the herbal floral quality of this aroma and therefore only detect the soapy note [77]. Additionally, the more copies of a specific olfactory receptor one has the more sensitive one will be to certain odorants. Moreover, the stronger an odor is perceived to be the more likely it is to be unpleasant, as regardless of the sensory system, high intensity stimuli are perceived as more unpleasant than the same stimuli at moderate intensity [78]. These issues imply that past experience and genetic variability need to be evaluated before adopting an odor-evoked memory regimen for a given individual.

### 4.3. Gender and Age

Women are generally more sensitive to odors than men are [79], and this enhanced sensitivity may influence the degree to which odor-evoked memories elicit physiological and emotional responses. Women have been shown to be more emotionally reactive to odors [80], and more susceptible to emotional conditioning with odors [81,82,83]. Lehrner, Eckersberger, Walla, Potsch and Deecke [84] also found that women, but not men, experienced less anxiety and a calmer and more positive mood when they were exposed to an orange aroma while awaiting an anxiety provoking event. However, in an extended replication of Lehrner et al. [84], Lehrner and colleagues [69] tested both orange and lavender aromas in comparison to a music and no-odor condition and found that both odors were able to improve mood and reduce anxiety equally among men and women. 

It should be noted that many studies examining odor-evoked memory have involved a preponderance of female participants (e.g., [13,14,32,36,37]), and sometimes only tested women (e.g., [18,33]). In descriptive studies that directly compared odor-evoked memories experienced by men and women it was observed that women described more emotional memories than men did [85,86]. However, other odor-evoked memory investigations involving experimental paradigms have not reported gender differences [31,87]. Gender also interacts with age with respect to how odors elicit memories. Zucco, Aiello, Turuani and Köster [88] investigated age and gender differences in odor-evoked memories, and found that among adolescents (12–14 years of age) and young adults (21–26 years of age), odors evoked more autobiographical memories among females than among males. However, with participants between the ages of 65–70 years, the effectiveness of odors to elicit autobiographical memories was equivalent between the genders—and, notably, the number of autobiographical memories generated to odors increased with age. The qualitative nature of odor-evoked memories as a function of age or gender was not examined in this study. In any event, generalizing the psychological and physiological results from odor-evoked memory research equivalently between men and women should to some extent be tempered, and the effects may also be mediated by personality factors. 

### 4.4. Personality

Personality can modulate the degree to which odors elicit emotional states. Devriese and colleagues [89] reported that neurotic individuals (a moody temperament, and a tendency towards anxiety and negative affect) were more likely to generalize acquired somatic symptoms, such as hyperventilation, in response to odors. There is also some evidence that emotionally labile individuals have greater absolute sensitivity to some odors [87]. Chen and Dalton [80] further found that women high in trait anxiety perceived hedonically polarized odors (odors rated as distinctly pleasant or unpleasant) more intensely than a neutral odor, and that men who were high in neuroticism or anxiety detected hedonically polarized odors faster than a neutral odor. 

In the specific realm of odor-evoked memory, high trait anxiety subjects were shown to feel more brought back in time, experienced higher arousal, and took slower, deeper breaths during the experience of odor-evoked memories than participants low in trait anxiety [35]. The general personality predisposition towards nostalgic reverie has also been found to correlate with experiencing more odor-evoked nostalgia [32]. In sum, as with “precision” medicine, the therapeutic implementation of odor-evoked memories should be tailored for the individual in question.

## 5. Conclusions

Mood improvement is desirable at numerous levels and this review has shown that the positive emotional states elicited by odor-evoked memories can ameliorate psychological and physiological responding and lower stress [35,36,43]. Moreover, through psychoneuroimmune interactions the emotionally positive qualities of odor-evoked memories appear to be able to reduce inflammation [36,37]. Inflammation is the basis of the majority of disease states and premature death [63]. Thus harnessing the power of positive odor-evoked memories has appreciable implications for benefiting health. 

Any odor that for a given individual evokes a happy autobiographical memory has the potential to increase positive emotions, decrease negative moods, disrupt cravings, lower stress and decrease inflammatory immune responses, and thereby have a generally beneficial effect on psychological and physiological wellbeing. Odor-evoked memories may also be able to stimulate specific emotions, such as self-confidence, motivation and vigor, and thus energize behavior as a function of the specific emotions that a given odor-evoked memory evokes. For example, an odor that triggers the memory of winning an important race could inspire all of these states and trigger positive physiological consequences. Moreover, the special resistance to retroactive interference that is fundamental to olfactory cognition can make a specific odor linked to a meaningful past personal event extremely reliable as a therapeutic agent. 

The potential therapeutic value of odor-evoked memory is not a testament to aromatherapy—the general proposition that various plant-based aromas have the ability to influence mood and wellness—which has mostly unsuccessfully waded into the therapeutic domain due to a lack of scientific rigor and confusion regarding the mechanisms involved [90]. Rather, when odors are capable of eliciting emotional and physical changes it is due to the emotions, memories and associations that have been linked to an odor through past personal experiences, which are then elicited when the odor is encountered, and the psychological and physiological responses connected to the odor are recapitulated [45,47]. It is in this way that odor-evoked memories have been empirically verified to alter emotional, mental and physical states and thus how they may be used in therapeutic applications. 

A methodologically sound understanding of the mechanisms that elicit beneficial responses from odor-evoked memories makes the implementation of odor-evoked memories a plausible adjunctive treatment for numerous conditions. For example, in addition to manipulations involving explicit odor-evoked memories, olfactory associations can also be used to implicitly alter behavior in a healthy manner, as Gaillet and colleagues have shown with food choices [91]. Future research should now use this empirical foundation to explore the precise parameters involved. Additionally, further research is needed to systematically investigate how odor-evoked autobiographical memories may be used therapeutically in conditions ranging from depression to addiction to systemic inflammation, both in the immediate case and in the long term. This work will also need to take into account issues of odor exposure, such as adaptation, habituation, trigeminal activation, and time of day, as well as a number of individual characteristics such as idiosyncratic experience, gender, age, genetics and personality. 

The uniquely emotional nature of odor-evoked memories brings our past back to us more viscerally and transportatively than any other type of memory experience. This exceptional feature of olfaction is witnessed most poignantly in the tragic case where a person suddenly loses their sense of smell in an accident and along with it their connection to themselves and others, their emotional wellbeing, and their overall quality of life [48,92,93]. From numerous perspectives it is evident that the autobiographical memories and emotional associations that are triggered by odors are essential to our psychological and physiological health.

## Figures and Tables

**Table 1 brainsci-06-00022-t001:** Methods and results in odor-evoked memory research with implications for psychological and physiological health.

Authors	Methods	Results
**Studies Reporting Predominantly Psychological Effects**		
Arshamanian et al., 2013 [10]	Familiar odors evocative of an AM were compared with memories elicited by verbal labels for those odors during fMRI scanning. Post-scanning, odors and labels were re-assessed and participants provided subjective evaluations of their memories.	Odors elicited more emotional and positive memories, and more activity in parahippocampal, amygdala and tempopolar regions than labels.
Miles and Bernsten (2011) [13]	12 familiar stimuli presented in verbal, visual and olfactory formats were compared for various qualitative dimensions of memory.	Odor-evoked memories were more positive, and more relevant to one’s life story than memories elicited by verbal or visual cues.
Reid et al. (2015) [32]	12 common scents were evaluated for nostalgic evocation and associated feelings.	Higher levels of odor-evoked nostalgia elicited higher levels of positive affect, self-esteem, self and social connectedness, optimism, and life meaning. Nostalgic scents elicited 3× more positive than negative emotions, and 2× more positive emotions than in a previous study examining music-evoked nostalgia.
Toffolo et al. (2012) [33]	An aversive documentary was paired with an olfactory, visual or auditory cue. One week later memories for the documentary were compared.	The olfactory condition elicited more arousing, detailed and unpleasant memories of the documentary than the auditory condition.
Willander and Larsson (2007) [20]	10 common scents were presented in odor-only, verbal-only, and odor + verbal format as memory cues and various experiential factors were assessed.	Odor-only cues elicited the most positive, emotional, and evocative memories.
Vermetten and Bremner (2003) [34]	Case reports of three individuals suffering from odor-evoked PTSD.	Odors specific to a given individual’s past trauma evoked PTSD flashbacks that were intensely negative, persisted for decades, and did not extinguish with time.
**Studies Reporting Predominantly Physiological Effects**		
Masaoka et al., 2012 [35]	A self-selected perfume that evoked an AM for each participant was compared to pleasant/neutral CFs.	Odors that evoked AMs promoted deeper, slower and more relaxed breathing compared to the CFs. More evocative memories and deeper breathing was observed among individuals who scored highest in trait anxiety.
Matsunaga et al. (2011) [36]	A self-selected perfume that evoked an AM for each participant was compared to a pleasant unfamiliar CF. Evoked emotions and associations were evaluated using rating scales. HR, SC, and plasma IL-2 levels in response to the odors were measured.	AM fragrances elicited more positive emotion, increased feelings of comfort and happiness, and decreased anxiety compared to the CF. HR decreased, SC increased, and IL-2 decreased after smelling the AM compared to the cf.
Matsunaga et al. (2013) [37]	A self-selected fragrance that evoked an AM for each participant was compared to two pleasant generic CFs. Emotions and memories evoked were evaluated. Plasma TNF-α, IFN-γ and IL levels in response to the odors were measured. PET scans of neurological activity during odor exposure were assessed.	6.5 times more feelings of nostalgia was experienced to the AM than CF. TNF-α and IFN-γ were decreased after smelling the AM. mOFC and precuneus/PCC were significantly activated during AMs, and activity in the mOFC and precuneus/PCC were negatively correlated with IFN-γ concentration. No IL changes were observed.
Sayette & Parrot (1999) [38]	Chronic smokers sniffed either a pleasant odor, an unpleasant odor, or a no-odor control when they experienced a cigarette craving.	Sniffing odors reduced cravings for cigarettes compared to the control condition. There was a trend for odors that evoked a memory to be the most effective at diminishing urges.

AM = autobiographical memory; CF = control fragrance; HR = heart-rate; SC = skin conductance; PTSD = post-traumatic stress disorder; IL = interleukin; TNF = tumor necrosis factor; IFN = interferon; PCC = posterior cingulate cortex; mOFC = medial orbitofrontal cortex.

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
