# Peer review of "The Role of Odor-Evoked Memory in Psychological and Physiological Health"

_brainsci, 2016, doi:10.3390/brainsci6030022_

Round 1

Reviewer 1 Report

This paper is a state of the art presenting the singular characteristics of odor-evoked memory and their influences on psychological and physiological health.

I found the article well written and well documented.

Specific comments

* Table 1: I think that the studies presented in this table should be better organized in the presentation. Indeed, in the text, the authors presented first the studies about the effects of odors on psychological parameters and then, the author presented the studies about the effects of odors on physiological parameters. So, I suggest organizing the Table as follows to be coherent with the text:

Psychological:

1- Arshamanian et al., 2013

2- Willander & Larsson (2007)

3- Matsunaga et al. (2011)

4- Matsunaga et al. (2013)

5- Miles & Bernsten (2011)

6- Reid et al. (2015)

7- Vermetten & Bremner (2003)

8- Toffolo et al. (2012)

Physiological

1- Matsunaga et al. (2011)

2- Masaoka et al., 2012

3- Sayette & Parrot (1999

* The study of Czerniawska et al appears in the Table but the study is described in the section 3.1 « odor perception » line 277 so I suggest to remove this reference of the Table.

* In the section entitled “OdorEvoked Memory and Physiological Responses”, lines 225-229 “Neuroimaging studies investigating brain–immune interactions have revealed that prefrontal regions, ….to exert a direct influence on immune responses.” This paragraph is not well positioned. Indeed, all the studies presented here examine the impact of odors on systemic inflammation. Please, place this paragraph in a well-suited place.

* Note that two sections are numbered in the same way.  Line 153 3- OdorEvoked Memory and Physiological Responses  and line 266 3. Issues for Consideration.  Thank you to correct this.

* In the conclusion, Lines 405-407, the author should add some references about the impact of non-attentively perceived odors on behavior (for example, olfactory priming Gaillet, M., Sulmont-Rossé, C., Issanchou, S., Chabanet, C., & Chambaron, S. (2014). Impact of a non-attentively perceived odour on subsequent food choices. Appetite, 76, 17–22 )

Author Response

  Specific comments

 * Table 1: I think that the studies presented in this table should be better organized in the presentation. Indeed, in the text, the authors presented first the studies about the effects of odors on psychological parameters and then, the author presented the studies about the effects of odors on physiological parameters. So, I suggest organizing the Table as follows to be coherent with the text:

Psychological:

1- Arshamanian et al., 2013

2- Willander & Larsson (2007)

3- Matsunaga et al. (2011)

4- Matsunaga et al. (2013)

5- Miles & Bernsten (2011)

6- Reid et al. (2015)

7- Vermetten & Bremner (2003)

8- Toffolo et al. (2012)

Physiological

1- Matsunaga et al. (2011)

2- Masaoka et al., 2012

3- Sayette & Parrot (1999

 * The study of Czerniawska et al appears in the Table but the study is described in the section 3.1 « odor perception » line 277 so I suggest to remove this reference of the Table.

Thank you for your comments on how to improve the Table. Your recommendations have been followed in the revision along with the comments suggested for improving the Table by Reviewer 1. 

 * In the section entitled “OdorEvoked Memory and Physiological Responses”, lines 225-229 “Neuroimaging studies investigating brain–immune interactions have revealed that prefrontal regions, ….to exert a direct influence on immune responses.” This paragraph is not well positioned. Indeed, all the studies presented here examine the impact of odors on systemic inflammation. Please, place this paragraph in a well-suited place.

This paragraph has now been moved to precede the sections discussing odor-evoked memory effects on systemic inflammation. See lines 233-237.

 * Note that two sections are numbered in the same way.  Line 153 3- OdorEvoked Memory and Physiological Responses  and line 266 3. Issues for Consideration.  Thank you to correct this.

These sections were numbered by the journal office and have now been corrected.

* In the conclusion, Lines 405-407, the author should add some references about the impact of non-attentively perceived odors on behavior (for example, olfactory priming Gaillet, M., Sulmont-Rossé, C., Issanchou, S., Chabanet, C., & Chambaron, S. (2014). Impact of a non-attentively perceived odour on subsequent food choices. Appetite, 76, 17–22 )

Inclusion of a discussion of the importance of implicit odor priming and the paper by Gaillet et al. (2014) has now been added. See lines 426-429. 

Reviewer 2 Report

Object:

Review of the paper: “The Role of OdorEvoked Memory in Psychological and Physiological Health”

Author: Rachel S. Herz.

Submitted to: Brain Sciences, 2016.

General comment:

This article examines a fascinating, original and  interesting topic: how odorevoked autobiographical memories contribute to modify mood inducing positive emotional states and how they can be used for ameliorating psychological and physical health.

I have found this paper well written, innovative and accurate. As a whole it has scientific relevance and contributes to the progress of the knowledge in the area. 

I think that it is suitable to be published on Brain Sciences pending few minor revision which are reported below.

Minor comments:

page 2, line 71-73.  To improve the reported passages  I would suggest the author to examine, W. Tham, RJ Stevenson and LA Miller (2009), Brain Research Reviews, 62, 109, and J. Plailly, et al. (2008), 28, 5257,  The Journal of Neuroscience, both studies on thalamic function in olfaction.

page 7, given that  MHC (major histocompatibility complex) or linked genes influence body odours (suggesting a link between olfaction and immune system), I wonder if the author would like to consider the seminal paper by Wedekind et al. (1995), Proceedings: Biological Sciences, 260, 245, on  MHC-Dependent Mate Preferences in Humans, to expand the reported issues.

page 2, please modify table 1 - which content is very hard to be followed -   to achieve a neat, readable presentation (e.g., provide left-right alignment, author-quotation alignment, reduce characters, create a frame whereas to put the contents, a.s.o.). Also, add the correspondent extensive meaning to the acronyms (e.g., HR, SC, TNF etc.)  used there.

page 5 line 157, change “classical” to “evaluative”

page 13 (references) line 484, change “Schall” to “Schaal” Abstract: I would suggest to change  (…) “This article is not a testament to aromatherapy, but rather illustrates how through the experimentally validated mechanisms of (…) to “This article illustrates how through the experimentally validated mechanisms of (…), as in the text there is just a minimal reference to aromatherapy (p.10, line 389).

Author Response

Minor comments:

page 2, line 71-73.  To improve the reported passages  I would suggest the author to examine, W. Tham, RJ Stevenson and LA Miller (2009), Brain Research Reviews, 62, 109, and J. Plailly, et al. (2008), 28, 5257,  The Journal of Neuroscience, both studies on thalamic function in olfaction.

Inclusion of thalamic involvement in attending to odors and references to these papers have now been added.  See lines 71-75.

page 7, given that  MHC (major histocompatibility complex) or linked genes influence body odours (suggesting a link between olfaction and immune system), I wonder if the author would like to consider the seminal paper by Wedekind et al. (1995), Proceedings: Biological Sciences, 260, 245, on  MHC-Dependent Mate Preferences in Humans, to expand the reported issues.

Thank you for bringing up the connection between MHC/immune function and olfaction. Although Wedekind’s paper is very important, I do not think that the body-odor/mate selection/MHC story fits into this paper’s focus on odor-evoked memory and health and how the section on page 9 (previously page 7) discusses experiments demonstrating that odor-evoked memory can alter the activity of inflammatory cytokines.  Therefore, to keep the paper more streamlined, I have elected to omit this avenue of discussion.

page 2, please modify table 1 - which content is very hard to be followed -   to achieve a neat, readable presentation (e.g., provide left-right alignment, author-quotation alignment, reduce characters, create a frame whereas to put the contents, a.s.o.). Also, add the correspondent extensive meaning to the acronyms (e.g., HR, SC, TNF etc.)  used there.

I apologize for the appearance of the Table. It was reformatted by the journal when the original MS was submitted. I have provided a new table, that hopefully will be re-formatted, and whose contents has been modified according to your comments and the comments of Reviewer 2.

page 5 line 157, change “classical” to “evaluative”. 

This has now been changed.

page 13 (references) line 484, change “Schall” to “Schaal” Abstract: I would suggest to change  (…) “This article is not a testament to aromatherapy, but rather illustrates how through the experimentally validated mechanisms of (…) to “This article illustrates how through the experimentally validated mechanisms of (…), as in the text there is just a minimal reference to aromatherapy (p.10, line 389).

The spelling of Schaal is now corrected.

The Abstract has been changed as suggested.

Round 2

Reviewer 1 Report

The author has considered the recommendations and she has made the requested corrections.

I think that this article can be accepted in the present form. 

Author Response

Thank you for accepting the MS in it's present form and for your insightful comments which greatly helped improve the original manuscript.

Reviewer 2 Report

thank you so much for having sent to me the revised version of the ms by Dr. Herz, submitted to Brain Sciences.

The ms has been significantly improved and now it is ready to be
published in Brain Sciences.
However, just a short note:
when I have submitted my first comments one suggestion to the author was lost during the copy and past procedure (I have realised this some days later).
It is very easy to be edited, so I am reporting this suggestion here
below asking you to kindly forward it to Dr. Herz for her consideration:

1. page 5 line 120-123: please shift the Matsunaga reference to the
next - more suitable - section (i.e.: Odor‐Evoked Memory and
Physiological Responses)

Author Response

Thank you for helping me clarify the psychological and physiological findings of Matsunaga et al. (2011a).

The study by Matsunaga et al (2011a) is discussed twice, both in the section on psychological responses and then again in the section on physiological responses, because this study found both effects.  On lines 120-123, it is explained that: "Matsunaga et al. (2011a) found that when men and women aged 21-38 were presented with the scent of a perfume that was personally reminiscent for them, more positive emotion, increased moods of comfort and happiness, and a decrease in anxiety were experienced compared to when a pleasant fragrance that did not elicit personal memories was presented." 

These findings refer to psychological responses so I believe they should be kept here.  

However, on lines 124-127 discussion of the HR and SC responses obtained in Matsunaga et al. (2011a) were made and this has now been taken out and kept to the section on physiological responses only, where a comment from the psychological section has now been added. See lines 188-189.